# 🫐 Beyond Textual CoT: Interleaved Text-Image Chains with Deep Confidence Reasoning for Image Editing

## Abstract

Image editing with natural language has gained significant popularity, yet existing methods struggle with intricate object intersections and fine-grained spatial relationships due to the lack of an explicit reasoning process. While Chain-of-Thought (CoT) has been explored to enhance reasoning, purely textual CoT or CoT augmented with coordinate information is fundamentally limited in its ability to represent intricate visual layouts and lacks the necessary visual cues to guide the generation of fine-grained, pixel-level details. To address these challenges, we propose **Mu**ltimodal **R**easoning **E**dit (**MURE**), a novel framework that *shifts the visual editing process from purely text-based reasoning to a series of interleaved textual and visual rationales*. Our framework performs image editing using a natively multimodal, interleaved text-image CoT. This approach generates a step-by-step chain of reasoning where a textual description is followed by a corresponding visual cue, such as a positional mask that defined intended edited regions or a representation of new content. Furthermore, to mitigate the hallucination phenomenon of large language models, we introduce **M**ulti**m**odal **D**eep **C**onfidence (**MMDC**) reasoning paradigm. This paradigm explores a tree of visual reasoning paths at each step. By pruning low-quality branches using a deep confidence score from a reward model, it ensures the model consistently follows a high-quality trajectory towards the final edited result. The proposed method decomposes complex editing tasks into interdependent sub-tasks, achieving greater precision at each stage and yielding high-fidelity edited results. We define the formulation for interleaved text-image chains and release the first CoT-Edit-14K dataset, comprising 14K high-quality editing examples. Extensive experiments show that our method yields significant improvements across three image editing benchmarks, establishing a more effective reasoning framework for visual editing.

## 1 Introduction

Image editing with natural language instructions (Liu et al., 2025a; Li et al., 2025a; Comanici et al., 2025; Shi et al., 2024) has become increasingly popular, which eliminates the need for manual masks that are typically required by traditional methods (Guo & Lin, 2024; Lin et al., 2024). Traditional image editing systems primarily rely on diffusion-based models (Batifol et al., 2025; Feng et al., 2025; Bazyleva et al., 2025) to directly transfer textual instructions into visual components without an explicit reasoning process. These approaches struggle with complex cases involving intricate object intersections and fine-grained spatial relationships, that humans naturally encounter in daily life. This limitation significantly restricts their practical application potential. Meanwhile, Chain-of-Thought (CoT) reasoning (Wei et al., 2022) has emerged as a powerful tool for multimodal Large Language Models (MLLMs) (Mitra et al., 2024; Zhang et al., 2023b; Zheng et al., 2023; DeepMind, 2025), enabling them to tackle complex tasks (Guo et al., 2025b; Zhang et al., 2023b; Li et al., 2025b; Zhang et al., 2024b) by explicitly generating intermediate thought processes.

Motivated by this, prior work has attempted to leverage CoT for image editing from different perspectives. Some researchers leverage purely textual CoT to analyze instructions, which guides the inference of the final edited image (Kang et al., 2025; Deng et al., 2025). However, this approach

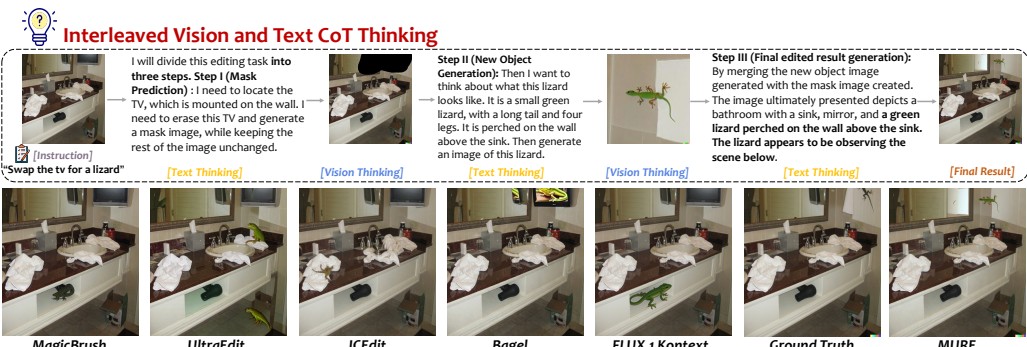

Figure 1: Visualization of our interleaved text-visual reasoning process and a comparative result. Given the prompt "swap the tv for a lizard", the MURE model **correctly performs multi-step reasoning to remove the lizard and its reflection in the mirror, generating a final edited image that maintains physical consistency**. In contrast, baseline approaches fail to handle this complex editing tasks, leading to erroneous results.

is fundamentally limited in providing precise control over object attributes and spatial relationships. To address this, other works, such as Generation of thought (GoT) (Fang et al., 2025; Duan et al., 2025), augment the textual CoT process with explicit coordinate information, typically in the format of bounding boxes. While these methods can manage the placement of visual elements, their CoT process remains inherently text-based, and is thus inadequate for representing complex visual layouts with intricate object intersections or irregular shapes. This, coupled with their inability to represent fine-grained visual details, can lead to unintended or inaccurate edits. As shown in Figure 1, existing approaches often fail to precisely localize edit regions and, consequently, produce physically inconsistent scenes. For instance, they may erase an object while leaving its reflection intact, thereby violating the laws of physics. This motivates our insight to alleviate this issue by introducing visual reasoning into the CoT process, as visual cues can provide precise intended edited regions, and also decomposing a complex task into several manageable sub-tasks, since high-quality completion of each step contributes significantly to the final high-quality result. Furthermore, focusing on the quality of intermediate steps is critical, as the inherent randomness of diffusion models (Zhang et al., 2023a) and the hallucination phenomenon of large language models (LLMs) (Zhang et al., 2025b; Bai et al., 2024) inevitably introduce low-quality elements that can degrade the final result.

Motivated by these insights, we propose **Mu**ltimodal **R**easoning **E**dit termed as **MURE**, a novel approach that *shifts the visual editing process from purely text-based reasoning to a series of interleaved textual and visual rationales.* Specifically, we formulate MURE as a multimodal reasoning chain that integrates both textual and visual information. Textual descriptions analyze instructions and provide powerful guidance for the subsequent reasonable image generation, while various visual components such as positioned masks for edit regions, representations of new content, or visualizations of intermediate actions provide explicit cues for final edited results. These textual and visual CoT processes are seamlessly combined to form an autoregressive, coherent, and well ordered chain that guides the model toward the final, satisfactory edited result. To enhance robustness within the inference stage phase, existing methods, such as parallel thinking or CoT with self-checking, have attempted to address this. However, parallel thinking can lead to inferior performance when low-quality traces dominate the voting process. As argued by Fu et al. (Fu et al., 2025), focusing on the local quality of the reasoning chains is far more important than concentrating solely on the final answer. To alleviate this issue, we propose a new paradigm, **M**ulti**m**odal **D**eep **C**onfidence (**MMDC**) reasoning. This method considers multiple reasoning paths at each visual generation step, forming a comprehensive reasoning tree. Based on this constructed tree, we prune low-quality branches of visual paths using a deep confidence score from a reward model. By selecting the most promising path, our approach enhances the robustness of the edited results and improves overall reliability.

As shown in Figure 1, in contrast to existing approaches, our proposed **MURE** model tackles such complex tasks by decomposing them into a series of interleaved textual and visual reasoning steps. This allows the model to first generate a precise intermediate state, such as the intended edited region that removes the lizard both in and out of the mirror. This foundational step ensures the generation of subsequent correct and physically plausible editing results. Ultimately, **MURE** step-

by-step approach, where generation is sequentially conditioned on all preceding outputs, allows the model to tackle intricate editing challenges and yield high-quality, satisfactory results. In general, our contributions are three-fold,

- We propose **MURE**, a novel approach that explicitly incorporates that employs a natively multimodal, interleaved text-image CoT to decompose editing tasks into a series of interdependent sub-tasks. This approach enables more precise subtask completion at each stage, thereby leading to higher-quality edited results.

- We propose the **MMDC** reasoning paradigm, which leverages a reward model's deep confidence score to prune low-quality visual paths, thereby enhancing the quality of intermediate reasoning steps and yielding high-fidelity edited results.

- We define the formulation for interleaved text-image chains and introduce the first **CoT-Edit-14K** dataset, comprising 14K high-quality editing examples to facilitate multimodal editing reasoning. Our experimental results demonstrate significant improvements across three image editing benchmarks.

## 2 RELATED WORK

**Image editing techniques.** Since the emergence of diffusion models, various image editing approaches have been proposed that can be broadly categorized into two types. Early works, such as training-free editing methods (Cao et al., 2023; Zhu et al., 2025), modify the original images through inversion in latent space and attention manipulation. Another line of work focuses on training-based approaches (Fu et al., 2023; Xie et al., 2025; Tong et al., 2024; Wang et al., 2024b; Chen et al., 2025a), which involve modifying model architectures or fine-tuning on high-quality datasets. For example, ICEdit (Zhang et al., 2025c) employs a LoRA-MoE hybrid tuning strategy to fine-tune the Flux.1 Fill model, while FLUX.1 (Batifol et al., 2025) leverages efficient flow matching technology in a novel latent space to achieve high-quality context-aware image generation and editing. More recently, models like Step-1X Edit (Liu et al., 2025b) and Seed Edit (Wang et al., 2025) leverage MLLMs to encode visual features and analyze instructions, which are then injected into a diffusion model. These approaches shows strong instruction-following and image-fidelity capabilities.

**Reasoning in Multimodal Large Language Models.** With the growing prominence of visual reasoning and MLLMs, CoT has been extended to both visual understanding (Yao et al., 2024; Xu et al., 2025) and generation (Guo et al., 2025c; Jiang et al., 2025; Chen et al., 2025b; Zhang et al., 2025a; Tong et al., 2025) tasks. Prior work has explored different forms of CoT: DeepSeek R1 (Guo et al., 2025a) uses a Vision-Language Model (VLM) to generate a comprehensive textual CoT for analyzing user instructions before generating a final answer, while MINI-CoT (Chen et al., 2025b) introduces visual tokens via cropping and zooming from original images to solve mathematical problems. MM-R1 (Liang et al., 2025) introduces a cross-modal CoT comprising visual components that spatially isolate concepts before generating the final image. For text-to-image generation (Liao et al., 2025), Uni-CoT (Qin et al., 2025) proposes a hierarchical design with separate CoT levels for task planning and subtask execution. ReasonGen-R1 (Zhang et al., 2025a) addresses this by introducing a large-scale CoT image generation dataset and integrating the CoT process into autoregressive image generation. However, the application of multimodal reasoning to image editing remains largely unexplored. Most existing editing approaches rely solely on textual CoT to analyze visual components, lacking the rich intermediate visual processes, such as explicit masks for edit regions or representations of new content, that are vital for synthesizing a high-quality final result.

## 3 METHODS

Most existing editing methods rely on **text-only reasoning**, occasionally supplemented with coordinate information. This approach is fundamentally limited by the inherent difficulty of representing complex spatial relationships and visual details through textual descriptions alone. To address these limitations, we propose **MURE**, a novel approach that *shifts the visual editing process from purely text-based reasoning to a series of interleaved textual and visual rationales.*

The overall framework of the proposed MURE model, illustrated in Figure 2, can be viewed as a sequential, natively multimodal CoT process. The model *addresses complex editing tasks by*

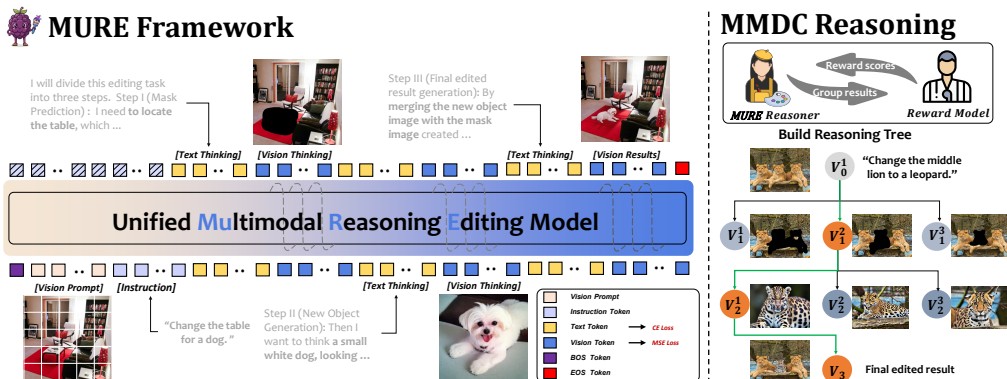

Figure 2: Overview of the **MURE** framework. **Left:** Our framework leverages an interleaved text-image CoT to achieve high-fidelity image editing. **Right:** The Multimodal Deep Confidence (**MMDC**) reasoning explores a tree of visual reasoning paths at each step. It prunes low-quality branches based on a deep confidence score from a reward model, ensuring a superior trajectory toward the final edited image.

*breaking them down into a series of interwoven textual and visual reasoning steps.* For a given task, such as "change the table for a dog", MURE first analyzes the user's instruction and the input image to plan the necessary subtasks, then executes them to generate the interleaved CoT step-by-step. The MURE model executes the editing task through the following steps: **Step I (Mask Prediction):** The model first analyzes the input image to identify and localize the object to be replaced. It then generates a textual CoT to describe the object's characteristics (e.g., "*a white footstool with a black frame*"). Conditioned on this text, the model generates a corresponding mask, effectively segmenting and removing the original object while preserving the rest of the image. **Step II (New Object Generation):** The model then forms a concept for the new object and generates a detailed textual description (e.g., "*a small white dog. Its fur is fluffy, and it is wearing a collar*"). The model's generation branch then renders a high-fidelity image of the new object that aligns with this textual description. **Step III (Final edited result generation):** This step involves leveraging the spatial information from the mask and the detailed content of the new object to ensure seamless and contextually appropriate integration. Simultaneously, the model generates a detailed description of the final edited image, using all this crucial information to produce a satisfactory edited result.

From this process, it is evident that the MURE model *decomposes a complex editing task into a sequence of manageable, interdependent subtasks, thereby enabling more precise subtask completion at each stage.* Implementing MURE requires three key components,

**A Comprehensive CoT-Edit-14K Dataset.** To facilitate interleaved text-image CoT editing, we introduce a comprehensive dataset. This dataset contains a collection of high-quality, carefully designed interleaved text-image CoT examples, each tailored to various editing subtasks. Such a dataset provides the foundational basis for enabling multimodal reasoning within the editing process.

**MURE: A Unified Framework for Visual Understanding and Generation.** The model is designed to understand and generate an interleaved natively multimodal CoT, enabling efficient inference of paired text and image outputs within a unified model framework. This unified architecture ensures that intermediate CoT representations are efficiently stored and retrieved using a Key-Value (KV) cache, which is crucial for maintaining state and coherence throughout the sequential generation process. Figures 12, 13 also demonstrate the **editable capabilities** of our interleaved CoT.

**MMDC: An Efficient Reasoning Paradigm for Interleaved CoT.** The inherent stochasticity of diffusion models often introduces low-quality intermediate visual steps, which can inevitably deteriorate the final result. To mitigate this, we explore multiple visual reasoning paths at each step, pruning low-quality branches with a deep confidence score to enable a more robust framework.

## 3.1 COT-EDIT-14K DATASET CONSTRUCTION

As illustrated in Figure 3, we propose a multi-stage pipeline to construct a comprehensive dataset of high-quality interleaved text-image CoT. This process is specifically tailored to each of the 10 unique

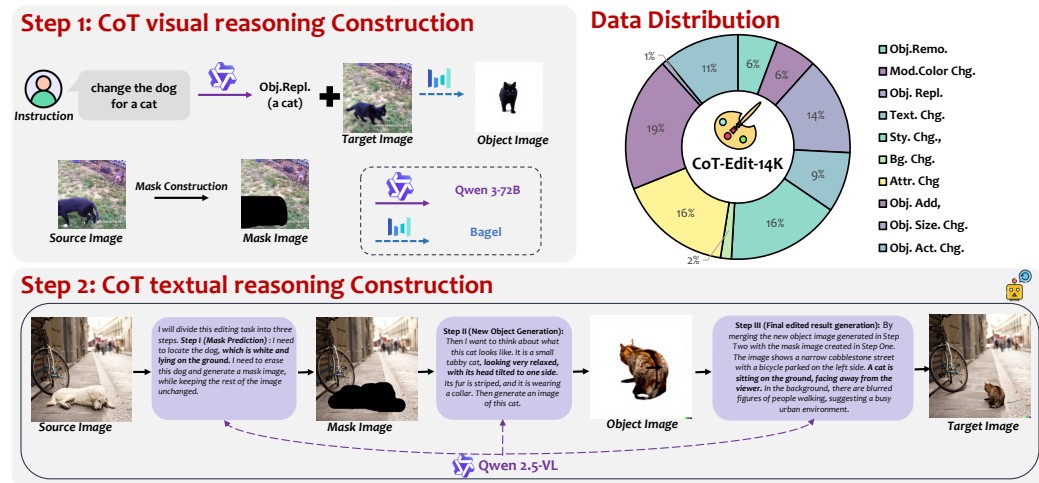

Figure 3: **MURE Dataset Construction Process. Top:** The visual annotation pipeline constructs explicit visual cues, including positional masks that define intended edited regions and valid representations of new content. **Bottom:** The textual annotation pipeline generates detailed textual descriptions based on the annotated CoT images from the top pipeline. The specific example illustrates the reasoning for an Obj.Repl. task, for detailed overview of the dataset, refer to Figure 7.

editing sub-tasks and involves two distinct pipelines, one for generating visual reasoning data and another for textual reasoning data. To ensure the quality of our dataset, we manually filtered out approximately 5K erroneous or low-quality examples from the raw data, resulting in 14K samples.

**For the CoT vision reasoning construction pipeline**, we first leverage Qwen-3 (Yang et al., 2025) to categorize editing tasks, as different tasks necessitate distinct CoT structures. We then follow Uniworld (Lin et al., 2025) to construct position masks using an adaptive editing region weighting strategy. Furthermore, we employ BAGEL with carefully designed in-context prompts to synthesize new objects or styles. For tasks involving motion, we also capture the intermediate process of motion changes from video data.

**For the CoT text reasoning construction pipeline**, we utilize Qwen2.5-VL (Bai et al., 2025) to generate the textual component of the interleaved CoT. This generation is conditioned on the editing instruction, the original image, and the CoT vision image from the previous pipeline. The completed text and image pairs are then seamlessly combined to form the complete CoT process.

We showcase our pipeline for constructing interleaved text-image CoT using the Object Replacement task as a representative example in Figure 3. Additional examples demonstrating the versatility of the MURE model across various tasks are provided in Figures 8, 9, 10 and 11 in the Appendix.

### 3.2 MURE FRAMEWORK

The implementation of an **interleaved text-image CoT** necessitates a model with unified multimodal understanding and generation capabilities. We thus adopt a unified model, such as BAGEL (Deng et al., 2025), as our backbone and finetune it for our specific task. The output of the designed **MURE** model, denoted $\mathbf{f}_\theta$, is a sequence of interleaved textual reasoning steps $s^{(i)}$ and visual tokens $v^{(i)}$, followed by the final edited image $O$. Specifically, $v^{(i)}$ comprises either a mask or new content depending on the editing step. The textual step $s^{(i)}$ and visual token $v^{(i)}$ are clearly delimited by the special tokens $\langle\text{visual start}\rangle$ and $\langle\text{visual end}\rangle$, respectively.

$$\left\{ s^{(1)}, v^{(1)}, s^{(2)}, v^{(2)}, \ldots, v^{(k-1)}, s^{(k)} \right\}, O \sim \mathbf{f}_\theta(\cdot \mid I, P) \tag{1}$$

where $I, P$ represent the input images and edit instruction, respectively.

**Training Objectives.** We train our model to generate text tokens autoregressively, optimizing the process with a Cross-Entropy (CE) Loss. This loss is computed at positions $\mathbf{T} \subset \{1, 2, \ldots, k\}$ that correspond to all textual segments $s^{(i)}$. Let $Y = \{y_1, y_2, \ldots, y_T\}$ denote the complete interleaved

text-iamge CoT sequence. The CE loss for next-token prediction is formally defined as,

$$\mathcal{L}_{\text{CE}}^{\text{text}} = -\sum_{t \in \mathbf{T}} \log P_\theta \left( s_t \mid y_{<t}, I, P \right) \tag{2}$$

For visual generation, we utilized Rectified Flow paradigm (Liu et al., 2022), Given a clean latent representation $z_0^{(i)}$ of the $i$-th visual image $v^i$, and a Gaussian noise sample $z_1^{(i)}$, the noisy latent variable $z_t^{(i)}$ is constructed by linear interpolation, defining a straight-line path between the two,

$$z_t^{(i)} = t \cdot z_0^{(i)} + (1 - t) \cdot z_1^{(i)}, \quad t \in [0, 1] \tag{3}$$

The model $f_\theta$ is trained to predict the velocity field on this straight-line path by minimizing the Mean Squared Error (MSE) loss. This loss is applied to the latent variables of all predicted visual segments within the sequence, including the final edited image, and is formally defined as,

$$\mathcal{L}_{\text{MSE}}^{\text{image}} = \mathbb{E}\left[\left\| \mathbf{f}_\theta\left(\mathbf{z}_t^{(i)} \mid y_{<t}, I, P\right) - \left(\mathbf{z}_0^{(i)} - \mathbf{z}_1^{(i)}\right) \right\|^2\right] \tag{4}$$

The context for our MURE model is the full preceding sequence, which comprises the original image, editing instructions, and a complete interleaved text-image CoT sequence, $Y = \{y_1, y_2, \ldots, y_T\}$, this sequence contains visual components, such as intended edit regions or the representations of new content, paired with their corresponding textual guidance. The overall objective for our training is a weighted combination of the CE and MSE loss functions.

$$\mathcal{L}_{\text{total}} = \lambda_{\text{CE}} \cdot \mathcal{L}_{\text{CE}}^{\text{text}} + \mathcal{L}_{\text{MSE}}^{\text{image}} \tag{5}$$

where the hyperparameter $\lambda_{\text{CE}}$ serves as a scaling coefficient to balance the relative contribution of the textual and visual objectives during training.

**Inference.** At the inference stage, given an initial image $I$ and an editing instruction $P$, the MURE model autoregressively generates an interleaved sequence of text and images $Y = \{s^{(1)}, v^{(1)}, \ldots, s^{(k)}, v^{(k)}, O\}$, by leveraging special tokens to flexibly switch between text and image generation. The process concludes automatically upon generating the final edited image $O$, which is signaled by a dedicated end-of-sequence token.

Compared to models that use purely CoT, such as the unified BAGEL model (Deng et al., 2025), our MURE framework takes a more sophisticated approach. It conditions the final edited result on a rich, interleaved text-image context, which is efficiently managed by a KV cache to ensure coherence throughout the sequential generation process.

### 3.3 MMDC Reasoning Paradigm

Low-quality intermediate visual steps may deteriorate the final result. Therefore, we introduce MMDC reasoning paradigm, which evaluates multiple reasoning paths and filter low-quality sub-path at each visual image generation step. Specifically, we leverage an independent reward model, $R_\theta$, which acts as an evaluator to score each candidate branch. This model is employed to assess the quality of each candidate visual image $v^{(k,i)}$ and its alignment with the paired textual instruction. We assign a deep confidence score, $S_{k,i}$ to each candidate image, which reflects its potential to contribute to a high-quality final output. This score is formally defined as:

$$S_{k,i} = R_\theta\left(v^{(k,i)} \mid s^{(k)}, y_{<k}, I, P\right) \tag{6}$$

In our interleaved text-image generation process, this score represents the model's confidence in a given reasoning path. We employ a greedy pruning strategy: at each visual generation step, we retain only the branch with the highest score and use it to guide subsequent text reasoning and visual generation. Ultimately, the optimal candidate image for the current step is selected by maximizing this deep confidence score. This selection process is represented by the following formula,

$$i^* = \underset{i \in \{1, \ldots, N\}}{\arg\max} S_{k,i} \tag{7}$$

Through this phased tree search, our model can effectively self-evaluate and correct errors during the generation process, selecting the path most likely to lead to a high-quality final output. This significantly enhances the robustness of the generated results. Note that different candidate branches in the reasoning process are executed in parallel. The reward model's prompt guidance is detailed in Figures 16 and 17, which includes **out-of-distribution (OOD)** generalization validation.

# 4 EXPERIMENTS

## 4.1 EXPERIMENTAL SETUP

We use BAGEL (Deng et al., 2025), a leading open-source unified model, to initialize our unified visual understanding and generation model. We trained this model on our CoT-Edit-14K dataset, which comprises approximately 14K interleaved text-image CoT pairs. To ensure the model retains its ability to perform single-step editing, we also incorporated data with text-only CoT, using system prompts to differentiate between the two modes. We utilize the Qwen2.5-VL-7B model as our reward model, eliciting its zero-shot capabilities through our specific prompt design to implement the guidance mechanism detailed in Figures 16 and 17.

## 4.2 EVALUATION SETTINGS

To validate the effectiveness of the proposed approach, we evaluate its performance on three benchmark datasets for image editing, MagicBrush test set (Zhang et al., 2024a), Emu (Sheynin et al., 2024), and SamrtEdit benchmark (Huang et al., 2024). To evaluate performance on the MagicBrush benchmark, we report a standard set of metrics utilized by prior works (Zhao et al., 2024; Zhang et al., 2025c). We report CLIP (Hessel et al., 2021; Shafiullah et al., 2022), DINO (Caron et al., 2021; Oquab et al., 2023), and L1 distance to measure the similarity between the edited images and their manually generated ground-truth (GT) images. Adhering to the evaluation protocols established by (Sheynin et al., 2024; Zhao et al., 2024; Zhang et al., 2025c), for the Emu edit benchmark, we employ CLIP-I and DINO to assess the similarity between the source and edited images, and CLIP-Out to quantify the distance between the generated output caption and the edited image. While for the SamrtEdit benchmark, to assess both visual fidelity and semantic alignment, we utilize PSNR, SSIM (Hore & Ziou, 2010), and LPIPS (Zhang et al., 2018) to measure perceptual consistency with the original image. We also calculate the CLIP Score (Hessel et al., 2021) to evaluate the semantic alignment between the edited image's foreground and the GT text label.

## 4.3 COMPARISONS WITH STATE-OF-THE-ART

Table 1: Quantitative results on **two benchmarks.** We compare our method against baselines on the **MagicBrush and Emu test sets**. Best results for each metric are highlighted in bold.

| Methods | CoT | MagicBrush Test Set | | | Emu Test Set | | |
|---|---|---|---|---|---|---|---|
| | | L1 ↓ | CLIP-I ↑ | DINO ↑ | CLIP-I ↑ | CLIP-Out ↑ | DINO ↑ |
| InstructP2P (CVPR23) | ✘ | 0.114 | 0.851 | 0.744 | 0.856 | 0.292 | 0.773 |
| MagicBrush (NeurIPS23) | ✘ | 0.074 | 0.908 | 0.847 | 0.877 | 0.298 | 0.807 |
| UltraEdit (NeurIPS24) | ✘ | 0.066 | 0.904 | 0.852 | 0.880 | 0.304 | 0.847 |
| FluxEdit (HuggingFace) | ✘ | 0.114 | 0.779 | 0.663 | 0.852 | 0.282 | 0.760 |
| FLUX.1 Fill (HuggingFace) | ✘ | 0.192 | 0.795 | 0.669 | 0.794 | 0.273 | 0.659 |
| RF-Solver Edit (arXiv25) | ✘ | 0.112 | 0.766 | 0.675 | 0.797 | **0.309** | 0.683 |
| ACE++ (arXiv25) | ✘ | 0.195 | 0.741 | 0.591 | 0.791 | 0.280 | 0.687 |
| ICEdit (arXiv25) | ✘ | 0.060 | 0.928 | 0.853 | 0.907 | 0.305 | 0.866 |
| Bagel (arXiv25) | Text | 0.067 | 0.923 | 0.856 | 0.869 | 0.308 | 0.824 |
| **MURE** | Text & Images | **0.049** | **0.943** | **0.877** | **0.920** | 0.301 | **0.897** |

Table 2: Quantitative comparison of different methods on Understanding and Reasoning scenarios across **SmartEdit** benchmark. The best results for each metric are highlighted in bold.

| Methods | Understanding Scenarios | | | | Reasoning Scenarios | | | |
|---|---|---|---|---|---|---|---|---|
| | PSNR↑ | SSIM↑ | LPIPS↓ | CLIP Score↑ | PSNR↑ | SSIM↑ | LPIPS↓ | CLIP Score↑ |
| InstructP2P (CVPR23) | 21.576 | 0.721 | 0.089 | 22.762 | 24.234 | 0.707 | 0.083 | 19.413 |
| MagicBrush (NeurIPS23) | 18.120 | 0.680 | 0.143 | 22.620 | 22.101 | 0.694 | 0.113 | 19.755 |
| InstructDiffusion (CVPR24) | 23.258 | 0.743 | 0.067 | 23.080 | 21.453 | 0.666 | 0.117 | 19.523 |
| SmartEdit-7B (CVPR24) | 22.049 | 0.731 | 0.087 | 23.611 | 25.258 | 0.742 | 0.055 | 20.950 |
| SmartEdit-13B (CVPR24) | 23.596 | 0.751 | 0.068 | 23.536 | 25.757 | 0.747 | **0.051** | 20.777 |
| Bagel (arXiv25) | 23.823 | 0.892 | 0.083 | 23.842 | 28.076 | 0.839 | 0.060 | 20.767 |
| **MURE** | **25.611** | **0.897** | **0.065** | **23.947** | **28.694** | **0.883** | 0.062 | **21.298** |

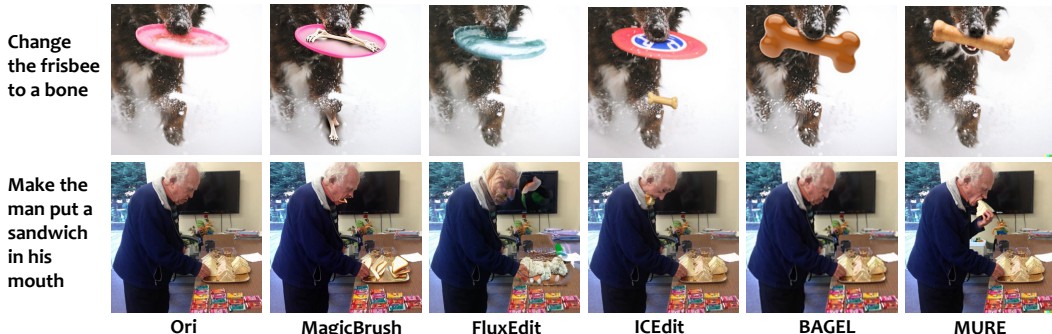

Figure 5: Visual comparison results. Additional examples are provided in Figures 14 and 15.

**Quantitative Results on Three Public Benchmarks.** We evaluate our MURE model against various approaches, including UNet-based (Brooks et al., 2023; Zhang et al., 2024a; Zhao et al., 2024; Geng et al., 2024), DiT-based (Mao et al., 2025; Paul, 2025; Black-Forest-Labs, 2024; Wang et al., 2024a; Zhang et al., 2025c), autoregressive-based (Huang et al., 2024), and unified model approaches (Deng et al., 2025). Our performance is detailed in Table 1, and 2. The proposed MURE model achieves performance on par with state-of-the-art (SOTA) across all three datasets. On the MagicBrush dataset, our output demonstrates strong alignment with GT images. On the Emu test set, our model achieves SOTA-level text

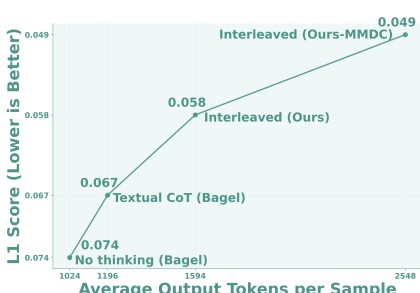

Figure 4: L1 Score vs. Output Token Cost on MagicBrush test set.

alignment while exhibiting superior image fidelity with original images. This strong performance is attributed to our approach's ability to predict intended edit regions within the interleaved text-image CoT. This explicit conditioning serves a dual purpose: it allows for precise regional modifications to mitigate editing errors and enhances background consistency by preserving unedited areas. Furthermore, on the SmartEdit benchmark, the proposed approach achieves superior performance compared to competing methods on both the understanding and reasoning scenarios, demonstrating robust performance on these complex tasks. This consistent strength is attributed to the application of the interleaved text-image CoT, which effectively breaks down complex problems into smaller sub-tasks. This multi-step process enables more precise task completion at each stage, allowing our approach to successfully accomplish highly intricate editing tasks. We also report L1 scores on the MagicBrush test relative to the average number of output tokens. Figure 4 illustrates a positive correlation between the model's performance and the combined textual and visual output token count, suggesting that a marginal increase in tokens yields a substantial improvement.

**Visual Comparison.** As Figure 5 demonstrates, the proposed MURE model achieves significant superiority over other open-source SOTA approaches in both editing accuracy and visual quality. This robust performance is attributable not only to our interleaved text-image CoT but also to the novel MMDC reasoning paradigm. The MMDC framework ensures the model efficiently follows a high-confidence inference path, effectively mitigating incoherent or sub-optimal outputs.

## 5 ABLATION STUDY

**Framework Design.** As shown in Table 3, we conducted an ablation study to analyze the impact of our framework's key components on the MagicBrush and Emu test sets. We started with a BAGEL model trained only with textual CoT. By incorporating interleaved text-image CoT, we significantly improved image quality, as evidenced by a 13.4% decrease in the L1 metric and a 1.3% increase in the CLIP-I score. This demonstrates that the MURE model's ability to decompose complex tasks into smaller sub-tasks leads to more precise task completion and a substantial performance gain. The introduction of the MMDC framework further enhanced our model's performance, particularly on the Emu test set. By leveraging a confidence score from a reward model

Table 3: Ablation study of our proposed MURE framework on MagicBrush and Emu test. The acronyms ICT and MMDC denote **I**nterleaved text-image **C**o**T** and **M**ulti**m**odal **D**eep **C**onfidence reasoning, respectively.

| ICT | MMDC | MagicBrush test | | | Emu test | | |
|:---:|:---:|:---:|:---:|:---:|:---:|:---:|:---:|
| | | L1 ↓ | CLIP-I ↑ | DINO ↑ | CLIP-I ↑ | CLIP-Out ↑ | DINO ↑ |
| ✘ | ✘ | 0.067 | 0.923 | 0.856 | 0.869 | **0.308** | 0.824 |
| ✔ | ✘ | 0.058 | 0.936 | 0.856 | 0.880 | 0.303 | 0.836 |
| ✔ | ✔ | **0.049** | **0.943** | **0.877** | **0.920** | 0.301 | **0.897** |

Table 4: Ablation study of our proposed MURE framework on MagicBrush and Emu test related to deep confidence reasoning search width.

| Search Width | MagicBrush test | | | Emu test | | |
|:---:|:---:|:---:|:---:|:---:|:---:|:---:|
| | L1 ↓ | CLIP-I ↑ | DINO ↑ | CLIP-I ↑ | CLIP-Out ↑ | DINO ↑ |
| 1 | 0.058 | 0.936 | 0.856 | 0.880 | **0.303** | 0.836 |
| 3 | 0.052 | 0.941 | 0.872 | 0.913 | 0.302 | 0.887 |
| 5 | **0.049** | **0.943** | **0.877** | **0.920** | 0.301 | **0.897** |

to guide visual image generation, this framework ensures the model follows a high-confidence path within the inference process, resulting in superior edited images.

**Search Width for Deep Confidence Reasoning.** Our deep confidence reasoning strategy uses a reward model to assess and prune intermediate image paths during interleaved text-image CoT inference. The results, as detailed in Table 4, demonstrate that increasing the search width (N) monotonically improves model performance. We hypothesize that employing a more capable reward model could further improve the overall effectiveness of our strategy.

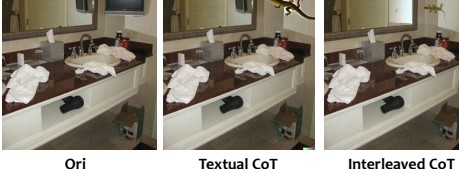

**Ori**      **Textual CoT**      **Interleaved CoT**

Figure 6: Qualitative comparison of our MURE framework, evaluating the impact of different reasoning chains on the test set. The editing prompt is "swap the tv for a lizard"

**Interleaved Text-image CoT vs. Textual CoT Reasoning.** As shown above, we enable our MURE model to perform image editing tasks using either a purely textual CoT or our proposed interleaved CoT. This is achieved by differentiating the reasoning process via a system prompt. In this subsection, we evaluate the editing performance of each method. The quantitative and qualitative results are presented in Table 5 and Figure 6, respectively. From these, we can observe that the interleaved CoT achieves better performance and generates more physically plausible results than the textual CoT. This also illustrates the superiority of decomposing a complex task into several manageable sub-tasks.

Table 5: Ablation study of our MURE framework, comparing textual vs. interleaved CoT on MagicBrush test set.

| Methods | L1 ↓ | CLIP-I ↑ | DINO ↑ |
|:---:|:---:|:---:|:---:|
| Bagel | 0.067 | 0.923 | 0.856 |
| Ours (Textual CoT) | 0.060 | 0.933 | 0.853 |
| **Ours (Interleaved CoT)** | **0.058** | **0.936** | **0.856** |

## 6 CONCLUSION AND LIMITATION

In this paper, we propose MURE, a novel framework that shifts the visual editing paradigm from purely textual reasoning to a series of interleaved textual and visual rationales. By decomposing complex editing tasks into manageable, interdependent subtasks, our approach enables more precise task completion. We demonstrate superior performance on three challenging benchmarks, MagicBrush, Emu, and SmartEdit, and attribute this success to two key innovations: an interleaved text-image CoT and our MMDC reasoning paradigm. The interleaved CoT enables fine-grained modifications and consistent background generation, while MMDC mitigates incoherence by ensuring a high-confidence inference path. Ultimately, MURE marks a significant step toward more effective, human-like reasoning for complex visual editing tasks.

ETHICS STATEMENT

Our datasets were meticulously curated from public sources to ensure they contain no personal information and comply with ethical guidelines. While our framework is designed for creative and beneficial applications, we explicitly acknowledge that its capability for high-fidelity synthetic editing raises serious concerns regarding the potential for misuse in generating misinformation or harmful content. We strongly condemn any such malicious application. The authors declare no conflicts of interest.

REPRODUCIBILITY STATEMENT

To ensure full reproducibility, we will make all code and data necessary to replicate our experiments publicly available upon paper acceptance. Comprehensive implementation details, including model architecture, hyperparameters, and training methodology, are provided in this paper and its appendix. We are committed to open-sourcing all essential resources to ensure that our findings can be fully verified and built upon by the research community.

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

# APPENDIX

## DETAILED SETTINGS

We initialize our MURE model using a pre-trained Bagel checkpoint and train it on our newly constructed CoT-Edit-14K dataset. This training enables the model to handle long, interleaved CoT sequences. To ensure it also retains its editing performance for purely textual CoT tasks, we incorporate training data pairs that use textual CoT to achieve image editing, with the two modes being distinguished via a system instruction. For the inference stage, we utilize Qwen2.5-VL-7B as the reward model. Training was conducted for 32,000 steps with a batch size of 1. The detailed training recipe is as follows,

Table 6: Training hyperparameters for the MURE model

| Hyperparameter | Value |
|---|---|
| *General Settings* | |
| Learning rate | $2 \times 10^{-5}$ |
| Optimizer | AdamW |
| Optimizer parameters | $\beta_1 = 0.9, \beta_2 = 0.95, \epsilon = 1.0 \times 10^{-15}$ |
| LR scheduler | Constant |
| Weight decay | 0.0 |
| Gradient norm clip | 1.0 |
| Training steps | 32,000 |
| EMA ratio | 0.9999 |
| # Training seen tokens | 0.24B |
| *Data & Model Dimensions* | |
| Max context window | 11600 |
| Sequence length per rank | (min: 10K, max: 14K) |
| Loss weight (CE : MSE) | 1 : 1 |
| *Generation Resolution* | |
| Generated image resolution | (min short side: 512, max long side: 1024) |
| Understood image resolution | (min short side: 512, max long side: 980) |
| Diffusion timestep shift | 4.0 |

## TASK ABBREVIATION EXPLANATIONS

Table 7: Explanation of image editing task abbreviations. The table also includes the average tokens for visual and textual CoT reasoning chains within our CoT-Edit-14K dataset. The utilized VAE has a $16\times$ downsampling factor. **Distinct from existing CoT editing datasets, we are the first to introduce interleaved text-image CoT for image editing.**

| Abbreviation | Full Term | Percentage | Avg. CoT Vision Tokens | Avg. CoT Textual Tokens |
|---|---|---|---|---|
| Obj.Remo. | Object Removal | 5.70% | 1024.0 | 165.72 |
| Mod.Color Chg. | Model Color Change | 5.97% | – | 104.16 |
| Obj.Repl. | Object Replacement | 14.15% | 1976.1 | 249.47 |
| Text Chg. | Text Change | 8.68% | 1328.3 | 114.40 |
| Style Chg. | Style Change | 16.43% | – | 79.23 |
| Bg.Chg. | Background Change | 1.56% | 1020.3 | 113.15 |
| Attr. Chg. | Attribute Change | 16.43% | – | 80.40 |
| Obj.Add. | Object Addition | 19.37% | 1078.3 | 141.19 |
| Obj.Size Chg. | Object Size Change | 0.74% | 1024.0 | 49.09 |
| Obj.Act. Chg. | Object Action Change | 10.97% | – | 110.22 |

SAMPLING FROM OUR PROPOSED COT-EDIT-14K DATASET

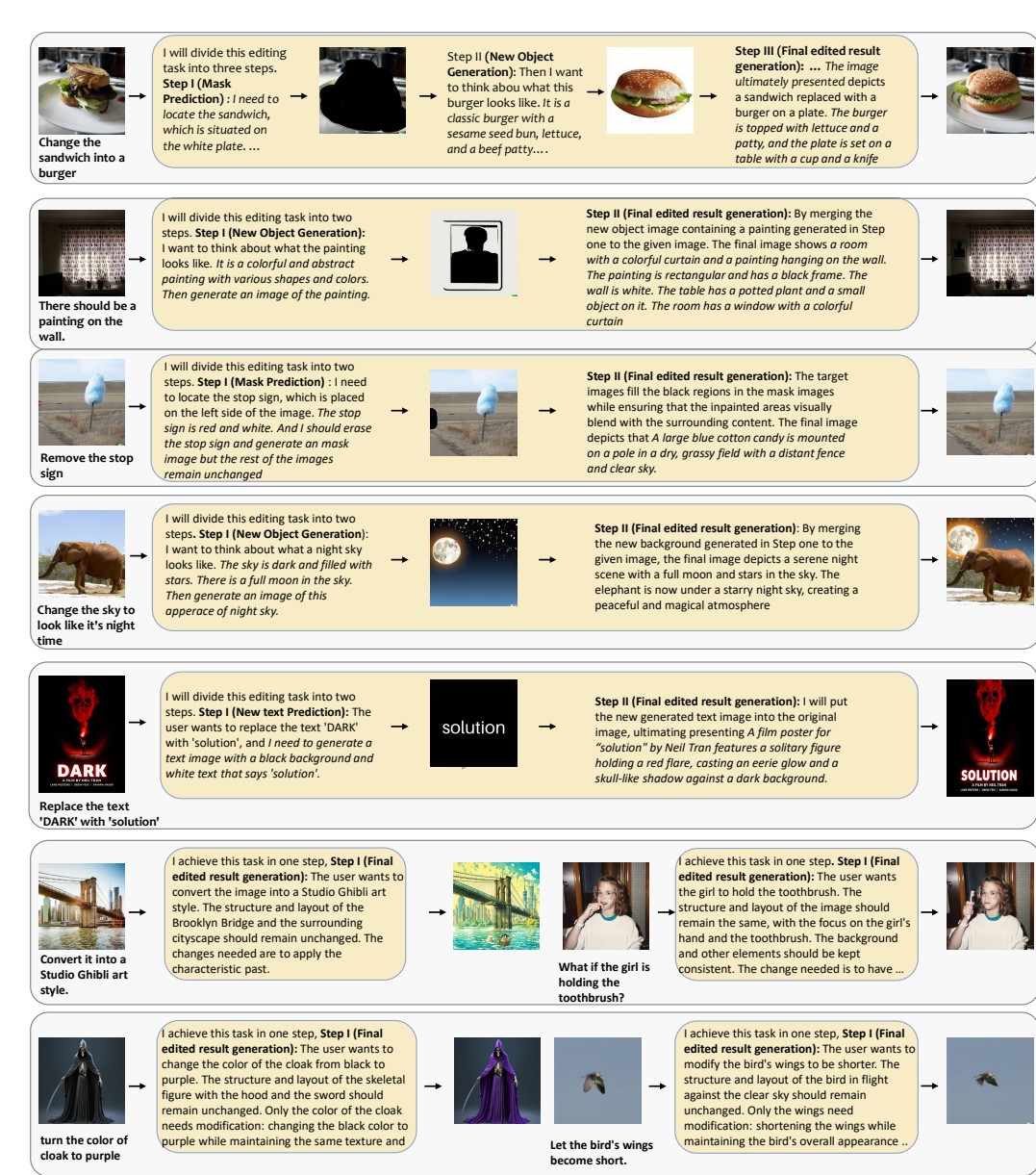

Figure 7: Examples from our CoT-Edit-14K dataset. The yellow highlights denote the CoT process, demonstrating distinct reasoning processes constructed for different types of editing tasks.

VISUALIZATION RESULTS FOR INTERLEAVED TEXT-IMAGE CHAINS FROM OUR MURE MODEL

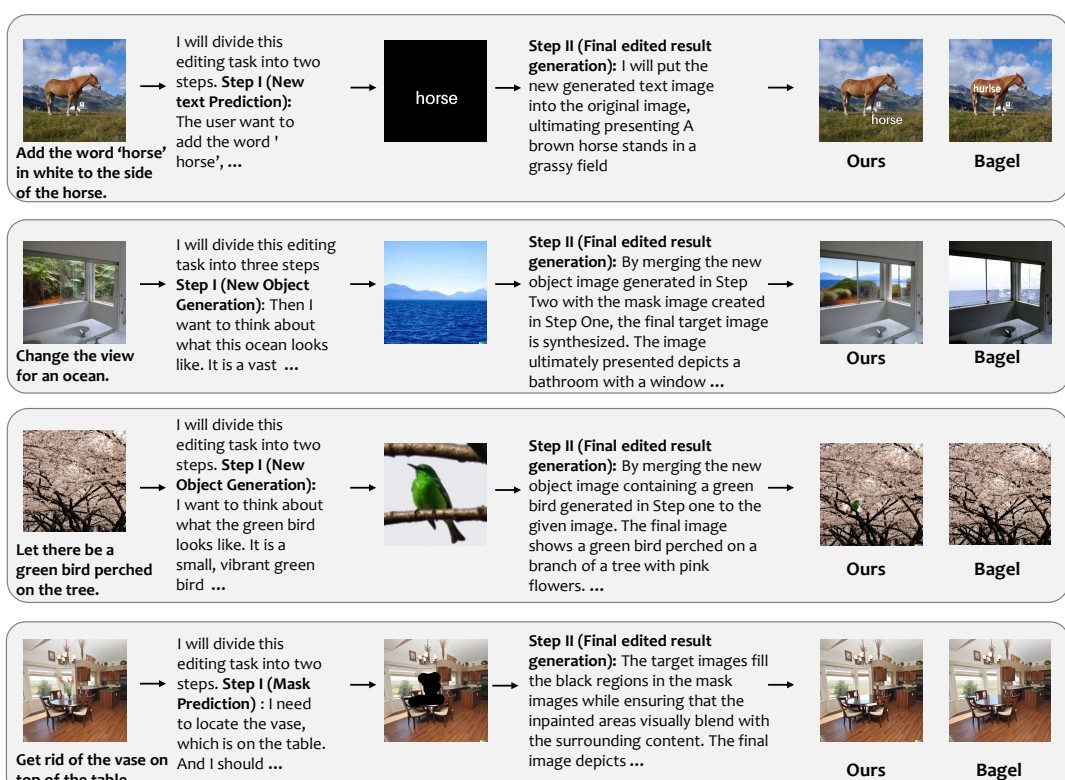

Figure 8: Qualitative visualization of MURE model on **a diverse range of editing tasks**, demonstrating its versatility and robust capabilities across various editing categories.

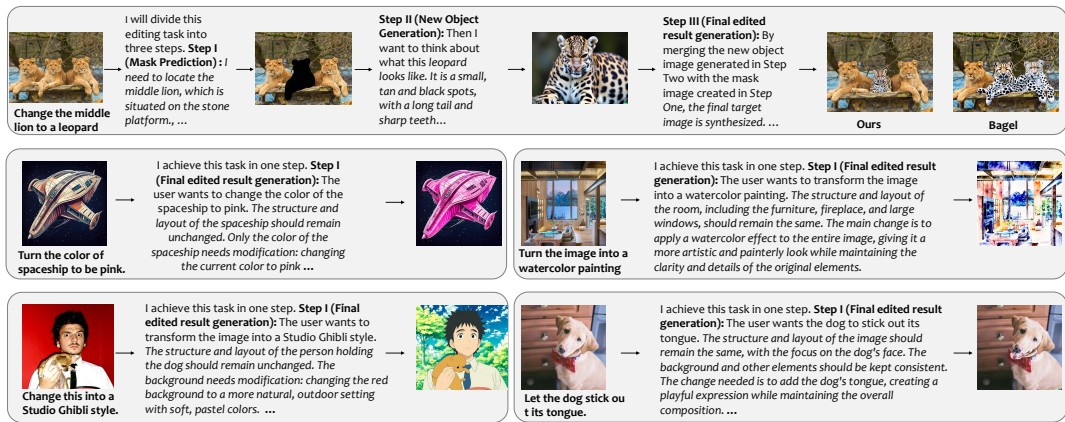

Figure 9: Qualitative visualization of MURE model on **a diverse range of editing tasks**, demonstrating its versatility and robust capabilities across various editing categories.

COMPARISON OF TEXTUAL AND INTERLEAVED CoT

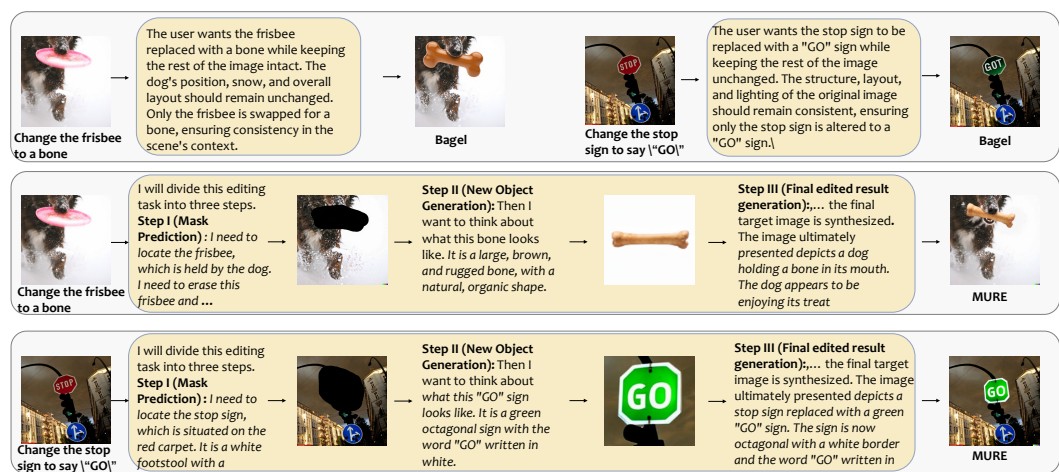

Figure 10: Qualitative visualization comparing purely textual and our interleaved CoT. The yellow highlighted regions illustrate the CoT process.

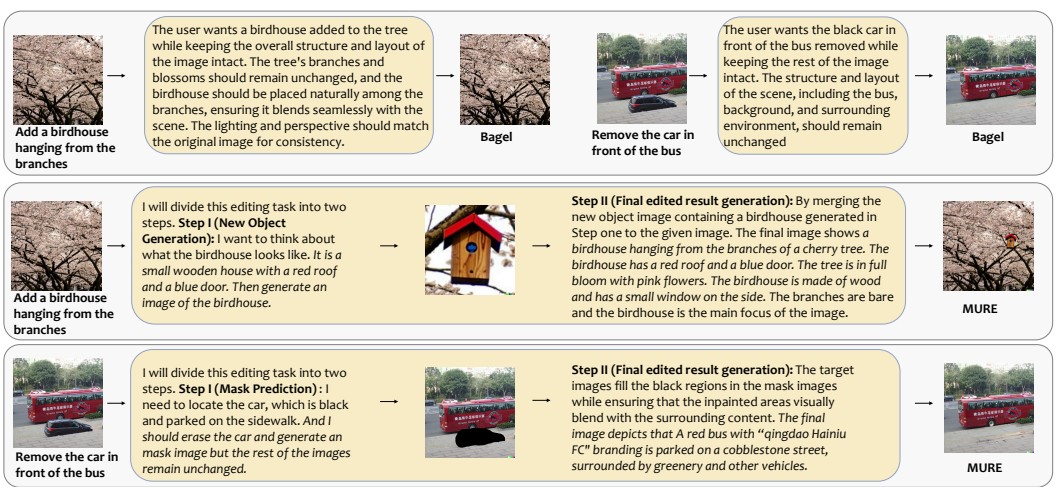

Figure 11: Qualitative visualization comparing purely textual and our interleaved CoT. The yellow highlighted regions illustrate the CoT process.

DEMONSTRATING THE EDITABLE CAPABILITIES OF INTERLEAVED COT

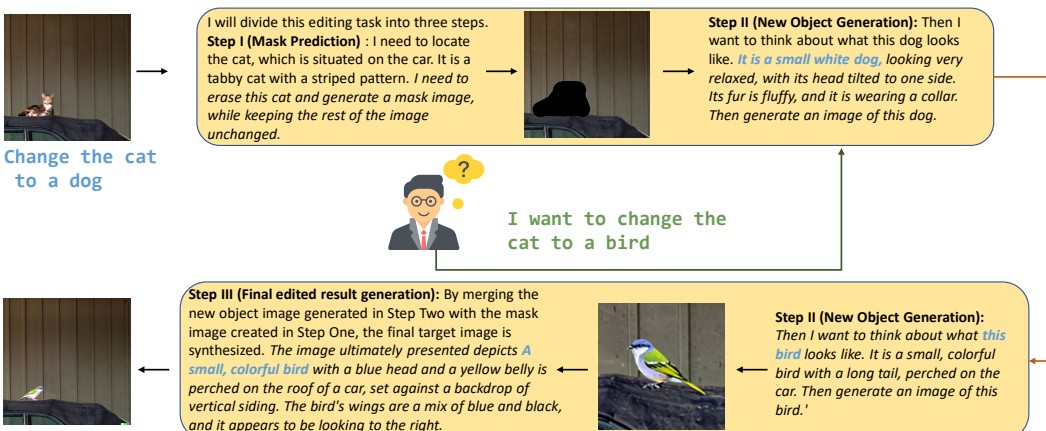

Figure 12: Our proposed MURE model demonstrates the **editable capability** of its intermediate CoT process. When given an initial prompt such as "change the cat to a dog", the model executes this editing task step by step. However, if the prompt is changed mid-generation to **"I want to change the cat to a bird", the MURE model can seamlessly adapt.** By leveraging the previously generated intermediate context, it continues to complete this new editing task without starting from scratch. This highlights the model's unique ability to handle dynamic instructions and reuse contextual information, a key advantage for interactive visual editing.

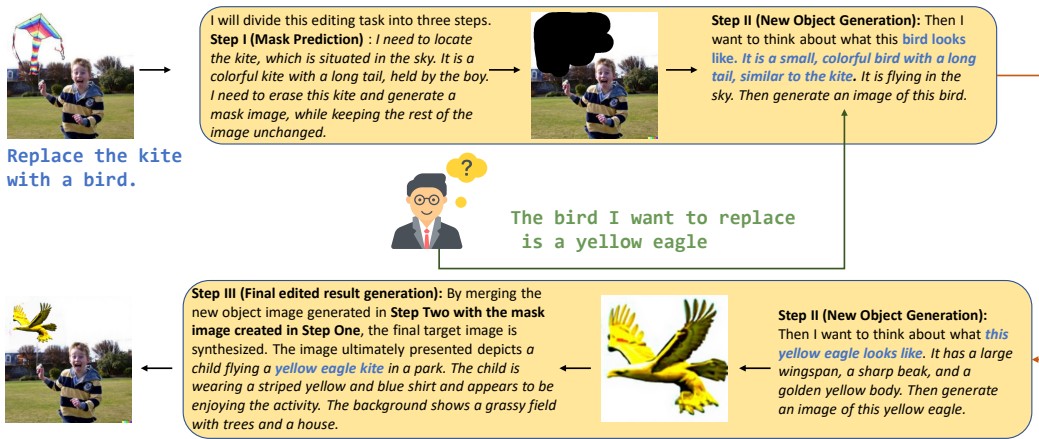

Figure 13: Our proposed MURE model demonstrates the dynamic and editable nature of its intermediate CoT process. When given an initial prompt like "Replace the kite with a bird", the model begins executing the task step-by-step. If the prompt is then updated mid-generation with more specific details, such as **"The bird I want to replace is a yellow eagle"**, the model can seamlessly adapt. By leveraging the previously generated intermediate context, it continues to complete the new, more specific task without re-initializing. This highlights the model's unique ability to handle dynamic instructions and reuse contextual information, a key advantage for interactive visual editing.

MORE QUALITATIVE RESULTS

Have the dog be wearing a backwards baseball cap

Change the stop sign to say \"GO\

Change the brown horse to white

Have there be a helicopter in the sky

Add a white horse in the front of the others

Add the word 'beach' to the sky.

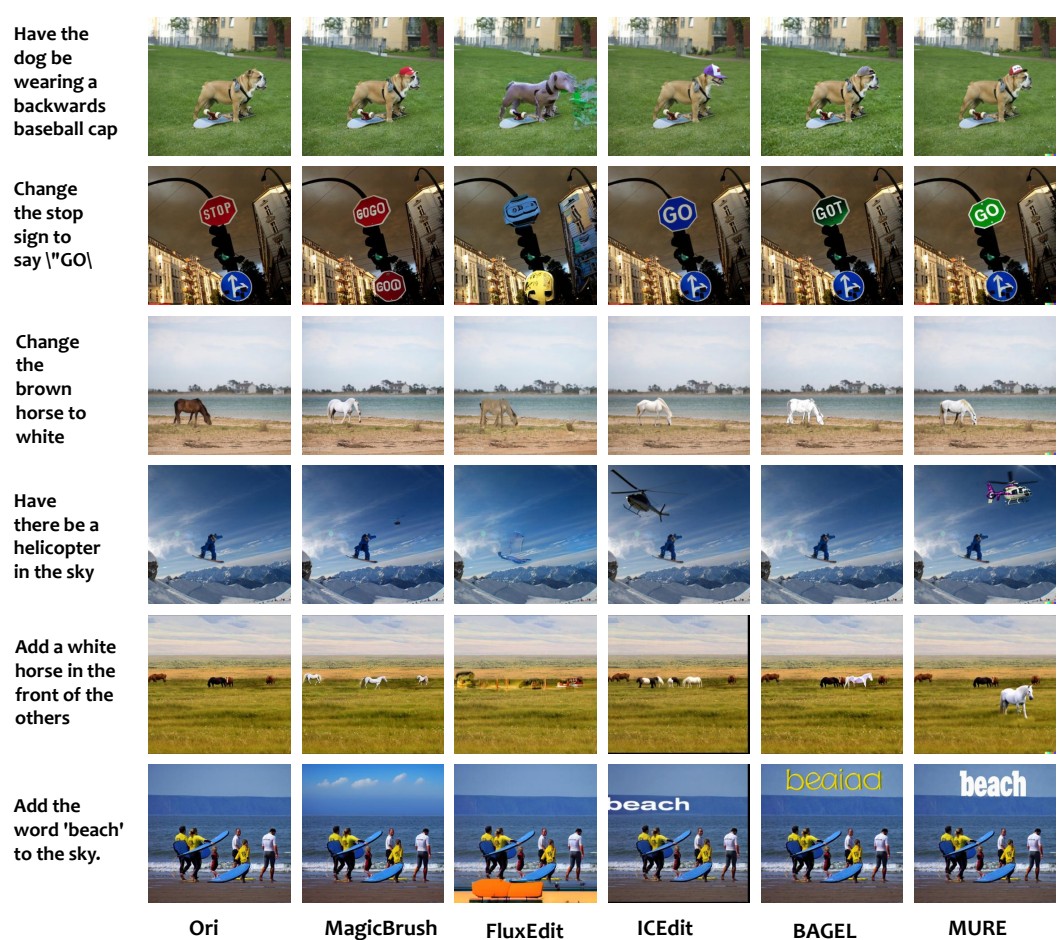

Ori    MagicBrush    FluxEdit    ICEdit    BAGEL    MURE

Figure 14: Qualitative visualization of MURE model on a diverse range of editing tasks, demonstrating its versatility and robust capabilities across various editing categories.

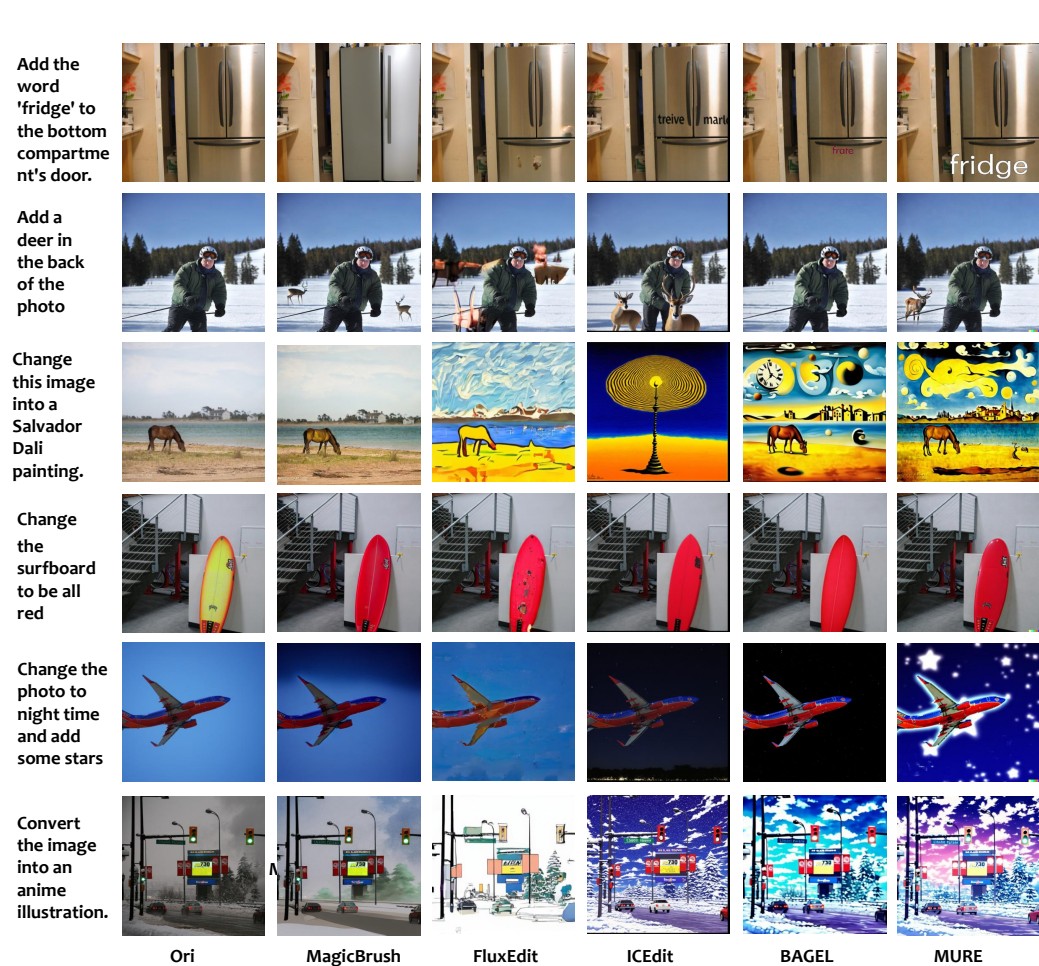

Figure 15: Qualitative visualization of MURE model on a diverse range of editing tasks, demonstrating its versatility and robust capabilities across various editing categories.

VISUALIZATION OF MMDC PARADIGM

> **Prompt guidance mechanism for the reward model:**
>
> **You are a professional image editing CoT instruction adherence ev aluator.** Your task is to evaluate a newly generated image and provide a single **Overall Score** from 0 to 100. This score should primarily be based on two criteria: **Instruction Adherence (70% weight)** a nd **Image Quality (30% weight)**.
>
> **Instruction Adherence** assesses how well the new image aligns with the **Chain of Thought Steps** and the original editing instruct ion, also with the original image (Note: you should evaluate the edited result according to the edited prompt and original images.). **Image Q uality** evaluates the clarity, naturalness, and absence of the new gene rated images.

Figure 16: Our reward model, Qwen-2.5 VL-7B, is leveraged for its zero-shot capabilities within our specific prompt guidance. The model is employed to evaluate a newly generated image by referencing its previously generated textual CoT and adhering to those aforementioned steps. We then score each candidate branch based on two dimensions: instruction adherence and image quality.

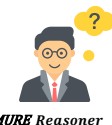
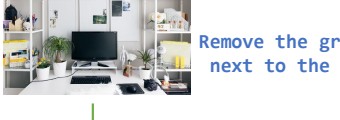
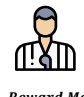

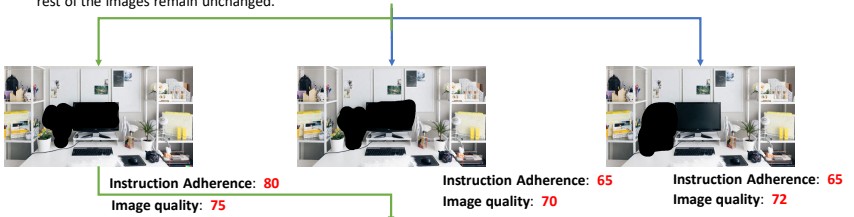

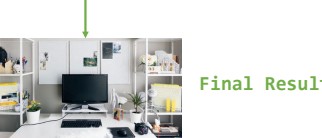

Figure 17: We demonstrate the **out-of-distribution (OOD)** capability of the proposed approach by applying it to an unconstrained, real-world photograph captured by a mobile device. The robust performance on this complex, user-generated sample is attributed to the **interleaved text-image CoT** and **MMDC** reasoning paradigm, which generates multiple visual reasoning paths at each step and selects the most promising one by leveraging a deep confidence score from a reward model, leading to a higher-quality final result.

VISUALIZATION OF PROMPTS IN DATASET CONSTRUCTION

**Prompt for Classifying Editing Instructions:**

Read one image editing instruction (**{prompts}**) and output exactly one line **like +<edited item>:** select the single best code or two code from *Obj.Add., Obj.Remo, Obj.Repl., Bg.Chg., Emo./Expr., Mod.Color Chg., Style Chg., Texture Chg., Obj.Move., Obj.Size Chg., Obj.Act. Chg., Persp./View Chg., or Others, then after the "+" write what will exist after the edituse the new action for Obj.Act. Chg. (e.g., meow), the new colour for Mod.Color Chg. (e.g., blue),* and for all other codes the object, background, style, etc.; output nothing else.

**Output:**

**Prompt for Textual CoT Reasoning on Final Image Synthesis for Obj.Repl. Task.**

**As a text based reasoning expert, please generate only the corresponding text prompt (answer) within 50 word based on the template I provide.** The first image, the second image and the three images are the original image and the new generated text image, and final edited images, respectively, and the edited prompt is {prompt} (Obj.Replac task).

**Templates:** The reference images for the task are the fourth, fifth, and sixth images, the edited prompt is Change the middle bird to a penguin, the output answer is *Step III (Final edited result generation): The target image is then synthesized by merging the object image containing the penguin with a mask image where the middle bird has been erased, ultimately presenting a penguin stands between two seagulls on a rocky surface, with one seagull grooming its feathers and the other facing the penguin. The background is blurred, suggesting a coastal environment.*

**Output:**

Figure 18: Visualization of prompts in dataset construction.

**Prompt for generating new object image within CoT image reasoning:**

Given this image, please identify the **{new_obj}** and extract only that specific **{new_obj}.** The generated image has high clarity, with a white background, but the details of the main subject are consistent with the given image.

**Output:**

**Prompt for generating CoT text reasoning for Style Chg. Task:**

Given a triplet of inputs—(1) **a question image**, (2) **a question text prompt**, and (3) **an answer image**—the objective is not to produce the final answer or image. Instead, you must *generate a chain-of-thought (CoT) process that articulates the reasoning leading to the answer image.*
*The reasoning should include an analysis of what aspects of the question image must be modified versus what must be preserved in the answer image.* Crucially, the analysis must emphasize the need to maintain the original image's structure and layout.
This reasoning process should incorporate an understanding of the context, user intent, and relevant background knowledge. The generated output must be concise, **with a length around or shorter than 60 tokens.**
**Example Output:**
The user wants to change the background from a sunny garden to a snowy setting. The structure and layout of the pink unicorn with bubble details and sunglasses should remain unchanged. Only the environment needs modification: replacing green grass with snow and surrounding greenery with frosted, snow-covered plants while maintaining lighting coherence.

**Output:**

Figure 19: Visualization of prompts in dataset construction.

> **Prompt for Textual CoT Reasoning on Mask Prediction for Obj.Repl. task**
>
> **As a text based reasoning expert, please generate only the corresponding text prompt (answer) within 50 word based on the template I provide.** The first image and the second image are the original image and the mask image, respectively. where the subject to be edited in the mask image has been erased, and the edited prompt is {prompt} (Obj.Replace task).
> **Templates:** The reference images for the task are the third and fourth images, the edited prompt is change the table for a dog, the output answer is *I will divide this editing task into three steps. Step I (Mask Prediction) : I need to locate the table, which is situated on the red carpet. It is a white footstool with a black frame, holding several items on top. I need to erase this table and generate a mask image, while keeping the rest of the image unchanged.*
>
> **Output:**
>
> **Prompt for generating CoT text reasoning about new object generation for Obj.Repl. task**
>
> **As a text based reasoning expert, please generate only the corresponding text prompt (answer) within 40 word based on the template I provide.** The first image and the second image are the mask image and the new object image, respectively. And the edited prompt is {prompt} (Obj.Replace task).
> **Templates:** The reference images for the task are the third and fourth images, the edited prompt is change the table for a dog, the output answer is *Step II (New Object Generation): Then I want to think about what this dog looks like. It is a small white dog, looking very relaxed, with its head tilted to one side. Its fur is fluffy, and it is wearing a collar. Then generate an image of this dog.'*
>
> **Output:**

Figure 20: Visualization of prompts in dataset construction.

STATEMENT ON THE USE OF LLMS

The authors of this paper declare that an LLM was used **solely for language editing and refinement.** The purpose of this usage was to improve grammar, correct spelling, and enhance the clarity and flow of the text. We did not use the LLM to generate any core content of this paper, including research ideas, experimental results, or data analysis. All scientific contributions and claims are the original work of the authors. We confirm our full responsibility for the final content of the submission, ensuring its accuracy and adherence to all ICLR ethical guidelines.

