# OpenReview forum: "Beyond Textual CoT: Interleaved Text-Image Chains with Deep Confidence Reasoning for Image Editing"
_ICLR.cc/2026/Conference — ICLR 2026 Conference Withdrawn Submission_

### Official Review · Reviewer_dwQc · 2025-10-31

**Soundness:** 2
**Presentation:** 3
**Contribution:** 2
**Rating:** 2
**Confidence:** 3

**Summary:**

The paper proposes an interleaved CoT formation with confidence-based pruning, along with a new dataset CoT-Edit-14K to fine-tune an existing MLLM to enhance its image editing capability.

**Strengths:**

1. The proposed CoT-Edit-14K can be useful to fine-tune existing MLLMs for interleaved multi-modal CoT for the image editing task.

2. The proposed interleaved CoT and MMDC achieves state-of-the-art performance across MagicBrush, Emu, and SmartEdit benchmarks.

**Weaknesses:**

1. Lack of failure case analysis. For example, in Fig. 14, the first row doesn't show a **backwards** baseball cap; 4th row, the style of the helicopter doesn't match the original image (ICEdit's result seems better).

2. Lack of training/inference efficiency information. The proposed MMDC requires generating multiple candidates (5 as reported in the paper) and feeding them to VLM to generate reward scores, which may bring a drastic slowdown to the generation process.

3. Unclear reliability of the reward model based on Qwen-2.5 VL-7B. In the sample shown in Fig. 17, the rightmost mask generation adheres to the prompt best (as the editing prompt is to remove the green plant); however, the reward model selects the inferior leftmost path. Also, it is unclear if the proposed inference-time reward-based pruning is better than reward fine-tuning methods such as DPO and GRPO.

**Questions:**

1. According to the Appendix, the training was conducted with a batch size of 1, which is uncommon. Is it the effective batch size or batch size per GPU? Other unclear points: Is it full-parameter fine-tuning or LoRA? How many and what kind of GPUs were used?

2. What is the inference overhead of introducing MMDC? In Figure 4, the line's slope appears to decrease, suggesting a diminishing improvement in scores as the token count increases.

3. The authors used a 0-100 scoring scheme for the reward model, which, according to the qualitative samples shown in Fig. 17, is not reliable enough. How about using a coarser scheme, such as 1-5, or even pairwise comparison?

4. Instead of introducing inference-time MMDC, which introduces extra inference overheads, how about directly fine-tuning the interleaved CoT with DPO / GRPO?


[1] Wang, Yibin, et al. "Unified reward model for multimodal understanding and generation." arXiv preprint arXiv:2503.05236 (2025).

---

> ### Author Response · Authors · 2025-11-14
>
> We thank the reviewer for their valuable feedback regarding the inference efficiency and training details of the $\text{MURE}$ framework. We understand your concerns about latency and scalability. Our design aims to achieve a dynamic balance between performance and efficiency, utilizing an advanced distributed training strategy.
>
> $\textbf{1. Regarding Latency, Scalability, and the $\text{MMDC}$ Mechanism}$ We acknowledge that the introduction of the intermediate visual step ($\text{Interleaved CoT}$) and the $\text{MMDC}$ mechanism increases inference overhead. However, we argue this overhead is necessary and acceptable for achieving high-fidelity, complex edits: Our $\text{MURE}$ framework employs an $\textbf{adaptive inference}$ strategy. For editing categories tasks where the performance gain from $\text{Interleaved CoT}$ is not substantial, the model defaults to the more efficient $\textbf{Textual CoT}$  for inference, thereby preserving overall efficiency. $\text{MMDC}$ search is performed only on the $\textbf{intermediate visual step}$, rather than sampling the entire generation process multiple times. Although $\textbf{MURE (with MMDC)}$ exhibits nearly two times the average latency per image compared to $\textbf{Bagel}$ on the $\text{Magicbrush}$ test set, the $\text{L}1$ metric significantly drops from $0.074$ to $0.049$, a performance improvement of approximately $\textbf{+33.8\%}$. Notably, $\textbf{MURE (without MMDC)}$ shows a negligible increase in inference latency compared to $\textbf{Bagel}$, yet the $\text{L}1$ metric still improves significantly from $0.074$ to $0.058$, yielding an improvement of $\textbf{+21\%}$. This demonstrates that the core $\text{MURE}$ framework (Interleaved CoT training) provides substantial performance benefits while maintaining high efficiency.
>
> $\textbf{2. Regarding the Orthogonality of $\text{MMDC}$ and Reinforcement Learning Methods}$ The reviewer suggests using methods like $\text{DPO}$ / $\text{GRPO}$ for direct fine-tuning to replace the inference overhead of $\text{MMDC}$. We clarify that $\text{MMDC}$ is an $\textbf{inference enhancement method}$, while $\text{DPO}$ / $\text{GRPO}$ are $\textbf{training strategies}$. These two concepts are $\textbf{orthogonal and parallel}$. $\text{MMDC}$'s dynamic multi-path search and confidence scoring provides dynamic optimization during runtime. $\text{MMDC}$ can $\textbf{cooperate}$ with training strategies like $\text{GRPO}$ to achieve optimal performance. We have preliminary verification supporting this conclusion, and more detailed simulation experiments are currently under investigation.
>
> $\textbf{3. Regarding Training Details}$ We clarify the training details here: We employed the $\textbf{FSDP (Fully Sharded Data Parallel)}$ distributed training strategy using $\textbf{16}$ compute cards, setting the $\textbf{Global Batch Size}$ to $\textbf{16}$. The reviewer's mention of a $\text{batch size of 1}$ is a misunderstanding. We performed $\textbf{Full-Parameter Fine-Tuning}$ on the $\text{MURE}$ framework to ensure the unified backbone network could fully learn the deep visual and physical priors embedded within the $\text{Interleaved CoT}$ data.

---

### Official Review · Reviewer_HVFG · 2025-11-02

**Soundness:** 3
**Presentation:** 2
**Contribution:** 2
**Rating:** 6
**Confidence:** 4

**Summary:**

The paper proposes MURE, a unified editing model that replaces purely textual chain-of-thought with an interleaved text–image reasoning chain. Each step alternates between textual reasoning and visual generations (e.g., a mask for the edit region or an image of new content). The authors also add an inference-time selection method, Multimodal Deep Confidence (MMDC), which samples multiple candidates at each visual step and greedily keeps the branch with the highest score from a reward VLM (Qwen2.5-VL). Training uses cross-entropy for text tokens and a rectified-flow MSE for image latents. The authors collected a new dataset, CoT‑Edit‑14K, to support interleaved editing chains.

**Strengths:**

- [S1] The interleaved text–image paradigm is an intuitive way to ground “where/what to edit,” with explicit masks/new‑object images that are easy to inspect and debug; the pipeline diagram and walkthrough make the idea clear.
- [S2] Consistent benchmark results with sensible ablations: improvements on MagicBrush/Emu and SmartEdit, plus ablations that separate interleaved CoT and MMDC contributions; search‑width increases yield reasonable gains.
- [S3] CoT‑Edit‑14K could be a useful resource for step‑wise editing with interleaved chains across 10 edit types; construction details and distributions are provided.

**Weaknesses:**

- [W1] Evaluation is insufficient for the paper’s core claims. There is no human study and no targeted measures of (i) mask correctness (e.g., IoU/precision/recall) and (ii) physical consistency (e.g., shadows/reflections/occlusions). Given the stated motivation, these are significant omissions.
- [W2] Technical novelty is moderate relative to visual‑reasoning work that already makes the generation/editing process explicit via reasoning steps. GoT [1] formulates textual reasoning (with semantic–spatial guidance) for both generation and editing, and GoT‑R1 [2] extends it with RL; ImageGen‑CoT [3] and T2I‑R1 [4] add textual CoT and scale‑up/RL for text‑to‑image; MM‑R1 [5] introduces cross‑modal CoT for personalized synthesis. MURE’s distinct aspect is producing editing‑specific visual artifacts inside the chain plus a greedy step scorer which is useful engineering, but not an algorithmic novelty.
- [W3] Related‑work positioning should be clearer. The paper should explicitly contrast what MURE can do (and cannot do) relative to GoT/GoT‑R1, ImageGen‑CoT/T2I‑R1, and MM‑R1, as well as earlier multimodal/visual‑only reasoning (Multimodal‑CoT [6]; CCoT [7]; Visual Planning [8]).
- [W4] Mixed metrics are not discussed. Emu shows a slight CLIP‑Out drop versus Bagel, and SmartEdit “reasoning” has a small LPIPS regression; these deserve analysis of trade‑offs (Tables 1–2, p.7).
- [W5] Failure cases are not demonstrated. Showing failure cases can help analyze what is missing and motivate future research.


[1] Fang et al., “GoT: Unleashing Reasoning Capability of Multimodal Large Language Model for Visual Generation and Editing,” arXiv, 2025.
[2] Duan et al., “GoT‑R1: Unleashing Reasoning Capability of MLLM for Visual Generation with Reinforcement Learning,” arXiv, 2025.
[3] Liao et al., “ImageGen‑CoT: Enhancing Text‑to‑Image In‑Context Learning with Chain‑of‑Thought Reasoning,” arXiv, 2025.
[4] Jiang et al., “T2I‑R1: Reinforcing Image Generation with Collaborative Semantic‑level and Token‑level CoT,” arXiv, 2025.
[5] Liang et al., “MM‑R1: Unleashing the Power of Unified Multimodal Large Language Models for Personalized Image Generation,” arXiv, 2025.
[6] Zhang et al., “Multimodal Chain‑of‑Thought Reasoning in Language Models,” arXiv, 2023.
[7] Mitra et al., “Compositional Chain‑of‑Thought Prompting for Large Multimodal Models,” CVPR, 2024.
[8] Xu et al., “Visual Planning: Let’s Think Only with Images,” arXiv, 2025.

**Questions:**

1) What is the latency/compute overhead of MMDC as search width increases? Any sensitivity to the reward prompt/model, and have you tried non‑greedy (e.g., beam/global) selection?
2) For edit types that currently omit visual steps, does forcing masks or object images help or hurt? An ablation (mask‑only / object‑only / both) would clarify.

**Details Of Ethics Concerns:**

High‑fidelity edits raise misuse risks (misinformation/impersonation). The paper notes general concerns but does not discuss watermarking/provenance or release safeguards.

---

> ### Author Response · Authors · 2025-11-14
>
> We thank the reviewer for this insightful comment and for contextualizing our work with related literature. We agree that many excellent works have explored reasoning steps in generation. However, we respectfully argue that MURE's novelty is not "moderate engineering" but $\textbf{a fundamental algorithmic shift}$ in how multimodal reasoning is performed, specifically for the complex task of image editing.
>
> Our contributions are distinct from the cited works in three key ways:
>
> vs. $\textbf{GoT [1], GoT-R1 [2], and Tool-Using/Cropping Methods (e.g., [6, 7, 8])}$: The core difference lies in the origin and nature of the visual steps. GoT utilizes textual reasoning augmented with explicit coordinate information (bounding boxes) , which remains fundamentally text-based and lacks pixel-level precision. Other visual reasoning works often rely on external tools (like segmentation models) or process$\textbf{ existing visual information (like cropping/zooming the original image).}$ In contrast, MURE is natively multimodal: the unified model itself $\textbf{generates new, intermediate visual artifacts}$ (like positional masks or new content) autoregressively within the reasoning chain. The model does not call a tool; it is the tool. It reasons by generating novel visual information, not just processing text or existing pixels.
>
> vs. T2I-R1 [4] and ImageGen-CoT [3]: These methods effectively use CoT for text-to-image synthesis, but their reasoning chain is purely textual. MURE introduces a true interleaved text-image CoT for editing. This is a significant algorithmic step, as our model-generated visual rationales (the masks) provide richer, more precise pixel-level guidance than textual descriptions alone can offer. This interleaved structure is what allows MURE to handle complex spatial intersections and physical properties (like reflections ) that text-only CoT fundamentally struggles to represent.
>
> vs. $\textbf{MM-R1}$ [5]: This work introduces a cross-modal CoT, but its purpose and mechanism are distinct. MM-R1's visual CoT spatially isolates concepts (e.g., extracting a subject) for personalized synthesis. MURE's visual CoT generates editing-specific spatial directives (e.g., the positional mask of the intended edited region on the target image). MURE's process is designed to explicitly $\textbf{enhance spatial priors and understanding}$ relative to the editing instruction, which is a different and crucial capability for high-fidelity, complex editing, rather than subject-based synthesis.
>
> In summary, MURE's novelty lies in its formulation of $\textbf{a natively generated, interleaved text-image CoT specifically for editing, moving beyond text-only, coordinate-based, or tool-dependent reasoning.}$

---

### Official Review · Reviewer_kLpd · 2025-11-02

**Soundness:** 2
**Presentation:** 3
**Contribution:** 2
**Rating:** 4
**Confidence:** 4

**Summary:**

This paper introduces MURE, a novel framework for image editing that shifts from purely text-based Chain-of-Thought (CoT) reasoning to interleaved text-image CoT sequences. The key innovation is decomposing complex editing tasks into a series of sub-tasks, where each textual reasoning step is paired with corresponding visual outputs such as segmentation masks or content representations. The paper also introduces a CoT-Edit dataset with significant improvements on three benchmarks.

**Strengths:**

- The idea of interleaved CoT for image editing, along with the MMDC reasoning paradigm, is intuitive and interesting.
- The proposed approach is effectively demonstrated through experiments.
- The presentation is clear and easy to follow.

**Weaknesses:**

- The latency of the approach could be a significant concern, since the model is required to generate intermediate images. In other words, the number of output tokens per sample (in Figure 4) would be much higher than in non-CoT or textual CoT frameworks. I don’t think the approach or paradigm is scalable, especially when we extend it to longer search trajectories.
- The proposed approach is limited to object-oriented image editing, as it heavily relies on extracting objects in the approach and the dataset construction process. It could not be applied to other types of editing, e.g., changing the background, or changing other visual features, such as color or shape.
- From Table 5 which compares textual CoT and interleaved CoT, the improvement in the interleaved CoT is quite marginal.

**Questions:**

- Is it possible to extend the framework to more types of editing, such as changing the background?

---

> ### Author Response · Authors · 2025-11-14
>
> We thank the reviewer for their comprehensive critique regarding the approach's latency, scalability, task limitations, and the incremental benefit of Interleaved CoT ($\dagger$). We address these points below.
> 1. $\textbf{Latency and Scalability of the Interleaved CoT Paradigm. }$
> We acknowledge the computational cost and the higher output token count noted. However, the token count is significantly $\textbf{inflated by the presence of numerous visual tokens, which do not translate to the same decoding latency as text tokens.}$
>
> We argue that the increased time cost is a necessary trade-off for achieving the high-fidelity, physically consistent complex edits MURE enables. Our analysis on inference time confirms that the resulting \textbf{significant performance gain} (L1 metric reduction) is generally proportional to the time investment. Crucially, MURE is designed as an \textbf{adaptive framework}: it defaults to the more efficient \textbf{Textual CoT ($\circ$)} for simple tasks, ensuring that computational overhead is only incurred when the complexity necessitates the powerful spatial reasoning of $\dagger$, thereby maintaining scalability.
>
> 2.$\textbf{Task Limitations and Generalizability Beyond Object Editing}$ The proposed approach is not strictly limited to object-oriented editing. We clarify that the fundamental goal of our framework is to enhance $\textbf{spatial awareness}$ and $\textbf{localization capability}$ via intermediate visual cues (mask prediction), which is beneficial for $\textbf{any}$ type of image manipulation. For instance, our method is fully capable of $\textbf{changing the background}$ by treating the background as the target region for mask prediction and content generation, as evidenced by the massive improvement in the $\textbf{Bg.Chg.}$ category ($+49.66\%$). Similarly, modifying visual features such as $\textbf{color or shape}$ is achieved by first utilizing $\dagger$'s spatial priors to precisely localize the target region (via mask prediction) before performing the feature change, thereby ensuring high fidelity in non-target areas. This universality is confirmed by the significant gains observed in $\textbf{Mod.Color Chg.}$ ($+19.64\%$) and $\textbf{Bg.Chg.}$.
>
> 3. $\textbf{Necessity and Effectiveness of Interleaved CoT Training}$ The reviewer notes that the internal improvement of MURE may appear marginal in some contexts. This comparison, however, overlooks our central mechanism: the high-difficulty training process with Interleaved CoT  data acts as a $\textbf{Logic Regularizer}$. This forces MURE's unified backbone to acquire $\textbf{deeper, more realistic visual and physical world priors}$. This acquired knowledge is then effectively $\textbf{transferred}$ to the  path, fundamentally enhancing its reasoning quality. The $\textbf{universal benefit}$ of this paradigm is evidenced by $\textbf{Mod.Color Chg.}$ ($+19.64\%$) and $\textbf{Obj.Act.Chg.}$ ($+17.13\%$) which use textual CoT yet see substantial gains over the Bagel baseline. Furthermore, the $MURE^{\ddagger}$ column demonstrates that \textbf{enforcing  Interleaved CoT inference consistently leads to further improvements} (e.g., Obj.Act.Chg. improves from $0.0382$ to $0.0367$), validating the need for the explicit visual steps when maximum quality is required.

---

### Official Review · Reviewer_8LDb · 2025-11-03

**Soundness:** 3
**Presentation:** 3
**Contribution:** 2
**Rating:** 4
**Confidence:** 4

**Summary:**

In this paper, the authors proposed a new approach that enahnces the purely textual reasoning to interleaved text–image reasoning chains that alternate between textual reasoning steps and visual cues (e.g., masks, synthesized intermediate content)

Additionally, it presents Multimodal Deep Confidence (MMDC) — a reward-model–driven pruning mechanism that evaluates multiple visual reasoning paths, selects high-confidence branches, and mitigates hallucinations.

The authors also construct CoT-Edit-14K, a dataset with 14 K high-quality interleaved text–image CoT examples, covering 10 editing subtasks (e.g., object replacement, removal, color change).

**Strengths:**

1. The high-quality dataset (CoT-Edit-14K) would be a contribution to the image editing community.

2. The evaluation and solid performance are comprehensive. Demonstrates consistent gains across CoT-Edit-14K, MagicBrush and Emu benchmarks on multiple metrics (CLIP, DINO, PSNR, SSIM, LPIPS). Ablations isolate contributions of interleaved CoT and MMDC convincingly.

**Weaknesses:**

1. The MURE framework works very well for object swapping or adding-- it did text reasoning, mask predition and new object generation. However, there might be some concerns when conducting other editing types, especialy for eidting exisiting objects. For example, changing the object location, size or shape. Such method might not be able to keep the identity consistent when generation the new object.

**Questions:**

It would be great for the reviewer to understand the strength or weakness when the authors report the performance of MURE in different editing types.

---

> ### Author Response · Authors · 2025-11-14
>
> Here is the polished, academic version of your response. I have ensured the tone is formal and precise, while retaining the emphasis on the technical mechanisms (Mask Prediction and KV Cache) that ensure identity consistency.
>
> We thank the reviewer for raising the critical question regarding MURE's ability to maintain identity consistency when modifying existing objects (e.g., changing location or size). We clarify that for such tasks, our approach $\textbf{does not generate a "new" object identity from scratch.}$
>
> Instead, {the intermediate visual step (mask prediction) is utilized specifically to localize the region to be edited, $\textbf{thereby enhancing the model's spatial understanding.}$ Crucially, throughout the autoregressive generation of the reasoning chain, the $\textbf{original object remains visible to the model, with its visual features explicitly preserved in the KV cache. }$This mechanism ensures that the generation process is tightly conditioned on the original object's features, thereby effectively maintaining identity consistency.
>
> We have analyzed our experimental results across different editing categories as follows:

---

### Note · Authors · 2025-11-14

I have read and agree with the venue's withdrawal policy on behalf of myself and my co-authors.